# Isolation and Characterization of Endomycorrhizal Fungi Associated with Growth Promotion of Blueberry Plants

**DOI:** 10.3390/jof7080584

**Published:** 2021-07-21

**Authors:** Binbin Cai, Tony Vancov, Hanqi Si, Wenru Yang, Kunning Tong, Wenrong Chen, Yunying Fang

**Affiliations:** 1College of Chemistry and Life Sciences, Zhejiang Normal University, Jinhua 321004, China; caibinbin@zjnu.edu.cn (B.C.); sihanqi37@zjnu.edu.cn (H.S.); yangwenru@zjnu.edu.cn (W.Y.); tongkunning@zjnu.edu.cn (K.T.); 2Zhejiang Provincial Key Laboratory of Biotechnology on Specialty Economic Plants, College of Chemistry and Life Sciences, Zhejiang Normal University, Jinhua 321004, China; 3NSW Department of Primary Industries, Elizabeth Macarthur Agricultural Institute, Menangle, NSW 2568, Australia; tony.vancov@dpi.nsw.gov.au

**Keywords:** blueberry, ericoid mycorrhiza, endomycorrhizal fungi, growth promotion, phosphate transporters *VcPHT1s*, transgenic *A. thaliana*

## Abstract

Despite their notable root mutualism with blueberries (*Vaccinium* spp.), studies related to Ericoid mycorrhizal (ERM) are relatively limited. In this study, we report the isolation of 14 endomycorrhizal fungi and their identification by fungal colony morphology characterization combined with PCR-amplified fungal internal transcribed spacer (ITS) sequence analyses. Six of the isolated strains were confirmed as beneficial mycorrhizal fungi for blueberry plants following inoculation. We observed the formation of typical ERM hyphae coil structures—which promote and nutritionally support growth—in blueberry seedlings and significant nitrogen and phosphorous content increases in diverse tissues. QRT-PCRs confirmed changes in *VcPHT1s* expression patterns. After the formation of ERM, *PHT1-1* transcription in roots was upregulated by 1.4- to threefold, whilst expression of *PHT1-3* and *PHT1-4* in roots were downregulated 72% and 60%, respectively. Amino acid sequence analysis of all four *VcPHT1s* genes from the blueberry variety “Sharpblue” revealed an overall structural similarity of 67% and predicted transmembrane domains. Cloning and overexpression of *PHT1-1* and *PHT1-3* genes in transgenic *Arabidopsis thaliana* plants significantly enriched total phosphorus and chlorophyll content, confirming that *PHT1-1* and *PHT1-3* gene functions are associated with the transport and absorption of phosphorus.

## 1. Introduction

Blueberries (*Vaccinium* spp.) have high nutritional and health values and contain considerable amounts of antioxidants such as anthocyanins and tannins [1]. In recent years, the scale of blueberry cultivation has greatly expanded in southern China due to the long growing season and its profitability [2]. The root system of blueberries is typically underdeveloped with small biomass and nonexistent root hairs, consequently necessitating the addition of cultivation substrates [3]. Although the blueberry industry has great potential to expand, the challenge of supplying adequate nutrients to the plant—thereby ensuring quality and yield—needs to be surmounted.

Phosphorus (P) is by and large one of the most essential macronutrients required for plant growth and development [4]. Inorganic P (Pi) in different types of soils is generally insufficient to meet the normal growth of plants owing to the low availability and mobility of P in soils [5]. Plants have evolved a variety of effective strategies to increase P absorption, namely mycorrhizal mutualism between fungi and plants [6]. Mycorrhizae posses two phosphorus transport pathways: one that directly absorbs via the root surface and root hairs in the soil interface, and the other absorbs by means of the hyphae in soil [7]. With either pathway, phosphorus transport proteins are quintessential in mediating uptake. Phosphate transporters are a class of carrier protein that actively transport phosphate and consist of two families, namely PHO and PHT transporters. The latter is divided up into three subfamilies—PHT1, PHT2, and PHT3 [8]. Most PHT1 family members are high-affinity Pi proteins located in the cell membrane and play an important role in transporting low concentrations of extracellular phosphorus into the cell under P limiting conditions [9]. 

The response of plant phosphate transporter genes to mycorrhizal symbiosis can be separated into mycorrhizal-inducible type and mycorrhizal-specific type. The expression of mycorrhizal-inducible phosphate transporter genes such as alfalfa *MtPT1*, *MtPT2*, *MtPT3*, *MtPT5* [10], and tomato *LePT3* [11] generally increases under low P conditions but diminishes in high P environments. Mycorrhizal-specific phosphate transporter genes are reportedly expressed only when the plant is inoculated with mycorrhizal fungi, for example as in alfalfa *MtPT4* [12], potato *StPT4* [13], and rice *OsPT11* [14]. Presently, the means by which ERM contributes to P uptake by plants, including their effect on inherent phosphate transport proteins, remains unclear.

Mycorrhiza is important for plant nutrient uptake, particularly for blueberries with poor root systems [15]. Endophytic mycorrhizas are classified as orchidaceous mycorrhiza (ORM), ericoid mycorrhizas (ERM), and arbuscular mycorrhiza (AM) based on the type of fungus and the location of fungal invasion [16]. Most mycorrhizas associated with blueberry roots belong to the ericoid mycorrhizas [16]. Although ERM is commonly found in wild blueberry varieties under natural conditions, they are relatively rare in cultivated blueberries [3]. Many mycorrhizal fungi have been confirmed to promote the growth and development of blueberries [16,17]. Inoculation of beneficial mycorrhizal fungi can enhance the biomass of blueberry cultivars to varying degrees [18,19]. Scagel found that mycorrhizas formed with the northern highbush blueberry variety called “Patriot” following inoculation with *Rhododendron* like mycorrhizal fungi can improve the efficiency of plant nutrient uptake and promote blueberry plant growth and development [20]. Some studies have demonstrated that the inoculation of ERM fungi on cranberry significantly increased the total nitrogen (N) content of the plants during different growth stages. This may because cranberries prefer ammonium nitrogen and that the accompanying ERM was able to mineralize organic nitrogen into ammonium nitrogen that could be absorbed in the medium, thereby facilitating plant N absorption [21]. Moreover, there are reports in the literature describing the presence of intra- and extracellular phosphatases in mycorrhizal fungi that promote the absorption of phosphorus by host plants [22]. Although there are several studies on the growth-promoting mechanism of ERM fungi in host plants [18,19,20,21,22], the means by which ERM fungi enhance blueberry growth remains uncertain and an attractive topic of research.

The study of ERM in blueberries is particularly worthwhile owing to potential economic benefits and because of its relative ease of use—viz that blueberries may be considered as model plant systems. In this paper, we examine and report the isolation and identification of mycorrhizal fungi from blueberries and evaluate their beneficial value in aiding blueberries accrue essential macronutrient elements in different tissues. Further, we explore the use of qRT-PCR to compare blueberry *PHT1s* gene expression levels in different tissues following inoculation with different beneficial endomycorrhizal fungi. Finally, we demonstrate and confirm the function of *VcPHT1s* genes via transformation and expression in *Arabidopsis thaliana*.

## 2. Materials and Methods

### 2.1. Plant Materials

Blueberry plant varieties used in this study were as follows: the southern highbush blueberry known as “Sharpblue”, which originated in Florida during the mid-1980s [23]; the multi-purpose “O’Neal” cultivar, which originated in the North Carolina [24]; and the “Premier” variety, which was released by North Carolina State University [25]. These blueberry varieties were selected for the study because they embodied good, medium, and poor growth pedigrees, respectively. For isolation of endomycorrhizal fungi, five-year-old blueberry plants (all three varieties) served as the point source. The southern highbush blueberry variety known as “Star”—released by the University of Florida as a patented variety in 1995 [25]—was used for experiments related to endomycorrhizal fungi inoculation and subsequent determination of physiological indexes. It was also used in studies investigating *VcPHT1s* (*PHT1-1*, *PHT1-2*, *PHT1-3* and *PHT1-4*) gene expression in roots and leaves. Further, this cultivar can avoid the specific preference of beneficial endomycorrhizal fungi isolated from the above three blueberry varieties. The “Sharpblue” blueberry cultivar was also used as a source for cloning of the *VcPHT1s* to further characterize their functions. All blueberry plant varieties were sourced from the Blueberry Planting Base in our laboratory, Jinhua, Zhejiang Province. Seeds of wild-type *A. thaliana* (ecotype Columbia) used throughout the study were derived from our laboratory.

### 2.2. Bacterial Strain and Plasmid Vector

*E**. coli* DH5α, *Agrobacterium tumefaciens* GV3101, the expression vector pCAMBIA1301, and pMD^TM^19-T vector were purchased from Shanghai Shenggong Biological Engineering Co., Ltd. (Shanghai, China).

### 2.3. Isolation and Identification of Blueberry Endomycorrhizal Fungi

Five-year-old “Sharpblue”, “O’Neal” and “Premier” blueberry varieties were used for the isolation of mycorrhizal endomycorrhizal fungi. Blueberry seedling root systems were collected, washed clear of soil, and air dried. Samples were surface-sterilized in the following manner: fibrous roots (divided into 2–3 cm root segments) were immersed in 75% ethanol for 30 s and rinsed several times with ddH_2_O to wash off residual ethanol. Roots were then rinsed in 10% sodium hypochlorite solution for 15 min and washed several times with ddH_2_O to remove residual sodium hypochlorite solution. For isolation and purification of endophytic fungi, twenty 0.5 cm root segments were individually inoculated into rose bengal [26] medium and incubated at 28 °C in the dark for 7 d. Following the appearance of hyphae on blueberry root segments, fragments of hyphae were aseptically transferred onto PDA solid media. Six individual fungal colonies from each plate were selected and streaked to purity onto PDA solid media plates and incubated at 28 °C in the dark for 7 days. Blueberry endomycorrhizal fungal isolates were morphologically characterized by visual examination of their structure, hyphae, and color, followed by molecular identification. Fungal total DNA was extracted using the CTAB method [27] and the ITS-specific regions (ITS1 and ITS2) adjacent to the fungal rDNA genes amplified using universal primers ITS1 and ITS4 [28]. The ITS PCR products were sent to Invitrogen (Shanghai) Trading Co., Ltd. for DNA sequencing. The ITS sequences were compared to sequences in the NCBI database using the Basic Local Alignment Search Tool (BLAST) program [29]. Bioinformatic analysis was performed to determine the species taxonomy of isolated fungal strains by constructing phylogenetic trees based on ITS sequences using maximum-likelihood analysis [30] with the MEGA7.0 software [31]. 

### 2.4. Screening of Beneficial Endomycorrhizal Fungi

For preparation of liquid inoculum, selected fungal isolates were inoculated into LB liquid media and incubated in the dark with shaking (200 rpm) for up to 3 days at 28 °C. At this point, the mycelium occupied approximately two thirds of the conical bottle and was applied as inoculum.

The southern highbush blueberry variety “Star” was used to screen endomycorrhizal fungi for their ability to bestow beneficial properties. One-year-old healthy and uniform “Star” plants propagated through tissue culture were separately transplanted into pots (one plant per pot with ordinary medium configured with a ratio of red loam:coconut bran:peat at 2:1:1). The seedlings were precultured for 15 d in a climatic chamber with day temperature at 27 °C for 16 h and night temperature at 20 °C for 8 h, with relative humidity maintained at 70%, light intensity of 400 μmol·m^−2^·s^−1^, and without any plant pruning. 

The study consisted of one control and seven treatments with three replicates for each: control plants were treated with conventional organic fertilizer (Jinhua Huijun bio-organic fertilizer) plus nutrient solution (see below), a treatment with CBF (commercial bacterial fertilizer, purchased from Jiangyin Lianye Biotechnology Co., Ltd., Wuxi, China), and the remaining six treatments inoculated with endomycorrhizal fungi isolates and fertilized with conventional organic fertilizer plus nutrient solution. A modified Hoagland nutrient solution [32] (Table 1) was used throughout the study and contains NH_4_^+^:NO_3_^−^ at a ratio of 4:1 (NO_3_^−^-N was from Ca(NO_3_)_2_, NH_4_^+^-N was from (NH_4_)_2_SO_4_), while the other nutrients K, Ca, Mg, and P are maintained at equal proportions. 

Prior to application around the blueberry seedling, 1 L Hoagland nutrient solution (hereafter referred to as nutrient solution) was added to 3 kg of organic fertilizer and mixed well. The pH of the nutrient solution was adjusted to 5.0 and sterilized by high-pressure steam. Approximately 20 mL of liquid inoculum was added to each plant, ensuring that it soaked down into the blueberry’s root system. All seedlings were moved to the climatic chamber with the same environmental conditions (as per above), watered twice a day (50 mL per plot), and nutrient solution reapplied monthly. After 3 months, blueberry seedlings were harvested for determination of physiological indexes and the appearance of blueberry seedlings photographed. Plant height is defined as the distance from the soil surface to the top of the plant and the crown growth as the radius of the plant’s crown, which was converted by the diameter.

### 2.5. Determination of Related Physiological Indexes

Root samples—first 2 cm—from the tip of the blueberry root system were harvested, washed clear of soil, and air dried. They were fixed with FAA [33] for 4 h and heated at 90 °C in 10% KOH for 1h. After rinsing in water, root fragments were decolorized in hydrogen peroxide, acidified with 5% solution of lactic acid, and stained with 0.05% trypan blue [34]. Prior to slide preparation, stained roots were rinsed with lactic acid and glycerin until the eluate was clear. Fungal colonization and morphology were examined using an inverted optical microscope (100× magnification). The number of infected root segments were counted under the inverted optical microscope (10× magnification) to quantify the infection rates by blueberry endomycorrhizal fungi using the root segment colonization weighting method [35]. Fully washed and dried roots of blueberry seedlings were numbered, and root activity was measured using the triphenyl tetrazolium chloride (TTC) reduction method [36].

Oven-dried roots, stems, and leaves of blueberry plants were collected and ground in liquid nitrogen (mortar and pestle). Approx. 0.05 g plant material was weighed out into a microwave digestion tube (corresponding to the digestion instrument) and mixed with 6 mL of concentrated sulfuric acid, followed by 4 mL of hydrogen peroxide (30%). Samples were subjected to microwave digestion (CEM MARS 6 Classic, Matthews, NC, USA) under the following conditions (i) 110 °C, 6 min; (ii) 150 °C, 8 min; (iii) 170 °C, 6 min; (iv) 140 °C, 5 min. N and P contents in the digestion samples were determined with Nessler’s reagent [37] and the ammonium molybdate spectrophotometric method [38], respectively. 

### 2.6. Quantitative Real-Time PCR (qRT-PCR) Analysis

Roots and leaves were quickly grounded in liquid nitrogen and total RNA was extracted from the roots and leaves of “Star” blueberries using the modified CTAB method [39]. Isolated RNA served as the template for cDNA synthesis with the PrimeScript II 1st Strand cDNA Synthesis Kit (TaKaRa Biotechnology, Dalian, China). Phosphorus transporter family genes—*PHT1-1*, *PHT1-2*, *PHT1-3*, and *PHT1-4*—DNA sequences from the bilberry database GDV (https://www.vaccinium.org/, accessed on 7 July 2020) were used in the design of qPCR primers (Table 2) with Primer Premier software v5.0 [40]. The qRT-PCR reaction system was composed of 1 μL cDNA, 0.5 μL forward primer, 0.5 μL reverse primer, 5 μL 2 × SYRB qPCR Mix, and 3 μL ddH_2_O. The expression level of individual samples was determined via qRT-PCR with the ABI StepOnePlus^TM^ Real-Time fluorescence quantitative PCR system (Applied Biosystems, Foster City, CA, USA) with three biological replicates and three technical replications. The glycerol-3-phosphate dehydrogenase gene (GAPDH) was used as the internal reference to normalize the relative expression levels of the target genes in quantitative experiments. The common 2^−ΔΔCt^ method was used to assess the relative expressions of *PHT1-1*, *PHT1-2*, *PHT1-3*, and *PHT1-4* genes in the roots and leaves of the blueberry plant under different experimental conditions

### 2.7. Cloning and Identification of VcPHT1s

The *VcPHT1s* genes were cloned from the “Sharpblue” blueberry variety for characterization and in preparation for creating transgenic *A. thaliana* plants. According to the bilberry database, *VcPHT1s* (*PHT1-1*, *PHT1-2*, *PHT1-3*, and *PHT1-4*) sequences were aligned using the CD-search in NCBI. Likewise, primers were designed by Primer Premier 5.0 for the full-length coding region of *VcPHT1s*, shown in Table 3. RNA was extracted from the leaves of “Sharpblue” blueberries and cDNA was synthesized using the method per Section 2.6. Full-length sequences of *VcPHT1s* were amplified using cDNA as the template with the PrimeSTAR^®^ Max DNA Polymerase (Takara, Wuxi, China) in a 50 μL reaction volume following the manufacturer’s protocol. The cDNA gene fragments were then ligated into the pMD^TM^19-T vector and transformed into DH5α *E. coli*-competent cells. Following screening, selected clones were sent to Instagram Trading Co., Ltd. for DNA sequencing. The DNA sequences were then subjected to the following bioinformatic analysis: the amino acid (aa) sequences of the proteins were compared using the DNAMAN software (Version 8, Lynnon Corp., Vandreuil, QC, Canada); phylogenetic trees of the aa sequences were constructed by the neighbor-joining method [30] using the MEGA 7.0 software package [31]; and the presence of transmembrane domains were predicted via TMHMM [41].

### 2.8. Plant Transformation and Function Analysis

The *VcPHT1s* genes were cloned into the vector pCAMBIA1301 producing the recombinants pCAMBIA1301-*PHT1-1*, pCAMBIA1301-*PHT1-2*, pCAMBIA1301-*PHT1-3*, and pCAMBIA1301-*PHT1-4*. The recombinant plasmids were introduced into *Agrobacterium tumefaciens* GV3101, which was subsequently used to transform *A. thaliana* by inflorescence soaking [42]. T1 seeds from the infested plants were screened for resistance on YEP solid medium containing Kan and Rif, and the integration and expression of *VcPHT1s* genes were confirmed by PCR amplification of isolated DNA and RNA. The seeds of T3 generation of *VcPHT1s* transgenic plants were sown in half-strength MS medium. *A. thaliana* plantlets with a minimum of three true leaves were then transplanted into coconut bran nutrient bowls containing three different P concentrations: namely 0 μmol/L, 500 μmol/L, and 1000 μmol/L. Phosphorous was supplied in the form of a modified MS nutrient solution [43] (removed organic composition) that contained KH_2_PO_4_-P instead of organic P.

After one month, plant growth of transgenic and wild-type *Arabidopsis* under different treatments were observed and leaf disc measurements, chlorophyll, and total phosphorus content were determined.

### 2.9. Statistical Analysis

The data were reported as mean ± standard error of three replicates. Single-factor analysis of variance was performed. Data were log-transformed to achieve normality where needed. The statistical significance was evaluated at the 0.05 probability level by Duncan’s new multiple range test.

## 3. Results

### 3.1. Isolation and Identification of Endomycorrhizal Fungi from Blueberry Plants

In this study, 14 fungal endophytes were isolated from the fibrous roots of three different blueberry varieties. Among them, 11 strains were isolated from “Sharpblue”, 2 from “O’Neal”, and 1 from “Premier”. According to ITS sequencing, 13 species belonged to Ascomycota, and 1 species (X5) was unknown but considered as an endomycorrhizal fungus due to its morphology. The colony morphologies of the isolates were predominately felted or in a flocculent formation, white to off-white in color with smooth margins, while a minority showed a radial pattern, secreted pink, yellow or green pigments (Figure 1A).

ITS DNA sequencing analysis of the 14 fungal isolates revealed that: (i) 6 strains, namely X2, A2, X6, X20, X29, and J4, belonged to the genus *Penicillium*; (ii) 4 strains (A4, X17, X22, and X28) were of the *Fusarium* genus; (iii) 2 strains, X24 and X11, belonged to *Gibberella* and *Aspergillus*, respectively; and (iv) the remaining 2 strains (X5 and X10) could not be classified (Figure 1B). Among them, isolate X11 was identified as the *Aspergillus* (HQ832844.1) with 100% bootstrapping, isolate X10 showed 100% bootstrapping to the unclassified fungi (GenBank: KC202937.1), while isolate X5 (unknown) was distantly related to other strains.

### 3.2. Inoculation with Beneficial Endomycorrhizal Fungi Significantly Increased the Growth of Blueberry Seedlings

Blueberry growth trials evaluating the benefits of *endomycorrhizal fungi* inoculation revealed significant differences between individual groups. Compared to the control, blueberry plants inoculated with CBF, X2, X6, X11, X24, A2, and A4 displayed more robust branches and the leaves were lusher with no evident chlorotic symptoms. Conversely, blueberry plants inoculated with strains X10, X20, X28, and J4 exhibited some negative effects such as seedling wilt, defoliation, and severe chlorosis (Figure 2A). 

The height and crown growth of blueberry seedlings inoculated with the 14 endomycorrhizal fungi presented varying effects compared to the control plants (Figure 2B,C). Blueberries inoculated with strains X2, X6, X11, X24, A2, and A4 increased plant heights by 17–41% and crown growth between 19–54%. Even compared to plants with commercial bacterial fertilizer, plant height and crown growth of seedlings inoculated by these six strains increased 6–28% and 7–38%, respectively. This confirms that fungal strains X2, X6, X11, X24, A2, and A4 are beneficial and promote blueberry growth. On the contrary, no significant growth (plant height nor crown growth) was evident in plants inoculated with the remaining strains. In fact, plant height and crown growth of seedlings inoculated with strains X10, X20, X28, and J4 were significantly lower (Figure 2B,C).

### 3.3. Infection Rates and Effect on Root Activities of Beneficial Endomycorrhizal Fungi in Blueberries

Following inoculation with beneficial endomycorrhizal fungi, two typical mycorrhizal colonization structures were observed in the roots of blueberry plants. One formed mycelial masses or hyphae coils within the plant cells (Figure 3A-I,A-II), while the other spread throughout the intercellular gaps between cells (Figure 3A-III). Of the latter, it appeared that some spread longitudinally or laterally throughout the intercellular and intracellular spaces of root cells forming complex networks (Figure 3A-IV). 

In contrast to the control, significantly higher infection rates (44–55%) were observed in plants inoculated with strains X2, X6, X11, X24, A2, and A4 (Figure 3B). Moreover, inoculation with endomycorrhizal fungi strains X2, X6, X11, X24, A2, and A4 generally increased root activities in blueberry seedlings (Figure 3C). Isolates X6, X11, and X24 were noteworthy because they increased root activities by 64–68% compared to the control, while A2 resulted in only a 31% increase.

### 3.4. Phosphorus and Nitrogen Content of Blueberries after Inoculation with Beneficial Endomycorrhizal Fungi

In comparison to control plants, total P content in roots, stems and leaves of inoculated blueberry plants was found to increase (Figure 4A,C,E). Inoculation with strains X2 and A4 led to a 41% and 48%, respectively, increase in leaf P content. Strains X2, X11, A2, and A4 similarly led to a 34–51% increase in stem P content. The P content in roots inoculated with these six beneficial endomycorrhizal fungi likewise increased significantly (15–47%). Generally, the N content of inoculated blueberry plant roots, stems and leaves also improved (Figure 4B,D,F). Compared to the control plants, the N content in the roots of seedlings inoculated with endomycorrhizal fungi significantly increased by 12–42%. Inoculation with strains X2 and X6 resulted in a 31% and 55% increase, respectively, in N content of stems. Besides X11, inoculation with all other strains resulted in a significant (19–24%) accumulation of N in the plant leaves. Interestingly, most inoculated plants had significantly higher P and N content in all three tissue samples than those treated with the commercial bacterial fertilizer. 

In summary, inoculation with all six strains increased the content of phosphorus and nitrogen in roots, particularly strains A2 and A4. Furthermore, the accumulation of phosphorus and nitrogen in stems and leaves improved considerably with the inoculation of strains X2 and A4. This implies that the growth-promoting effects ascribed to endomycorrhizal fungi are associated with the accumulation of phosphorus and nitrogen in different tissues of the blueberry plant.

### 3.5. Impact of Beneficial Endomycorrhizal Fungi on VcPHT1s Expression

Inoculation with X2, X6, X11, X24, A4, and A2 significantly affected the expression of *PHT1-1*, *PHT1-2*, *PHT1-3*, and *PHT1-4* genes in blueberry leaves and roots (Figure 5). Relative to uninoculated controls, transcription of *PHT1-1* in roots was upregulated by 1.4- to threefold higher following inoculation with isolates X2, X6, X11, X24, A2, and A4, while no significant difference was observed in the leaves of blueberry seedlings (Figure 5A). Expression of *PHT1-2* in inoculated blueberry seedling leaves increased by three- to 14-fold (Figure 5B). Although inoculation with these strains generally improved *PHT1-2* expression in roots (except for isolates X11 and A4), the levels were not as high. Apart from isolate X11, inoculated endomycorrhizal fungi generally stimulated inverse expression patterns for the two *PHT* gene variants in blueberry roots and leaves. Interestingly, inoculation with X11 led to a 20-fold increase in *PHT1-2* expression within the roots (Figure 5B). 

Expression of *PHT1-3* in roots was enhanced by 60–110% after inoculation with X2, X11, and A2, whereas A4 downregulated expression by 70% and 72% in leaves and roots, respectively (Figure 5C). Likewise, transcripts of *PHT1-4* in roots of the blueberry seedlings inoculated with A4 showed a significant decrease of up to 60%, whilst transcripts of other treated groups increased between two- and sevenfold. However, improvement of *PHT1-4* expression in the leaves was not significant, except for inoculation with isolate X2, which resulted in a 3.8-fold increase in expression (Figure 5D). Differential expression of *VcPHT1s* confirms that inoculation with beneficial endomycorrhizal fungi influenced observed phosphorus content (Section 3.4) in the different blueberry plant tissues.

### 3.6. Isolation and Bioinformatics Analysis of VcPHT1s

*PHT1-1*, *PHT1-2*, and *PHT1-3* were 1605 bp, 1617 bp, and 1617 bp in size, respectively, and encoded for approximately 550 amino acid proteins, while *PHT1-4* (1005 bp) protein encoded only 327 amino acids. The four amino acid sequences shared an overall structural similarity of 67.32%. The highest amino acid similarity was 77.9% between *PHT1*-3 and *PHT1*-1, while the lowest (25%) was between *PHT1*-3 and *PHT1*-4 (Figure 6). Analysis by the TMHMM software revealed that all four *VcPHT1*s contained transmembrane domains and that they were located on the plasma membrane, as predicted [9]. The phylogenetic tree containing the amino acid sequences of the four *VcPHT1s* and other reported *PHT1* aa sequences revealed that *PHT1*-1, *PHT1*-2, and *PHT1*-3 are in the same branch, while *PHT1*-4 is in a different branch (Figure 6B). *PHT1*-1 and *PHT1*-3 aa sequences displayed the closest relationship to a phosphate transporter from *Rhododendron williamsianum* (KAE9455466.1) [44], whilst the aa sequence of *PHT1*-2 was most similar to the phosphate transporter in *Actinidia chinensis* var. *Chinensis* (PSR94867.1) [45]. The *PHT1*-*4* aa sequence was closely related to the phosphate transporter in *Camellia sinensis* (XP_028106772.1—NCBI Annotation Release 100). 

### 3.7. Expression of VcPHT1s in Transgenic A. thaliana Plants under Different P Concentrations

The heterologous genes *PHT1-1* and *PHT1-3* were successfully introduced into *A. thaliana* by agrobacterium-mediated transformation. Transgenic Arabidopsis showed more vigorous growth and had more leaves than wild type (*wt*) plants with zero P (0 μmol/L) and low P (500 μmol/L) treatments, whereas no significant difference in growth was observed under normal P (1000 μmol/L) conditions (Figure 7A). 

Measurement of total P content, chlorophyll content, and leaf discs of transgenic *A. thaliana* and *wt* plants under different phosphorus conditions revealed comparable changes (Figure 7B–D). Under P stress (0 μmol/L and 500 μmol/L), overexpression of *PTH1-1 and PHT-2* significantly increased total P by 14–23% and chlorophyll content by 19–31% compared to the control. Moreover, leaf disc size of *PTH1s* overexpression in *A. thaliana* was between 3.8–5.7 cm, which was significantly larger than those in *wt* plants. However, under normal P treatment conditions, the differences between transgenic Arabidopsis and *wt* were smaller. These results demonstrated that *PHT1-1* and *PHT1-3* effectively responded to low P conditions, promoted phosphorus absorption, and alleviated symptoms of phosphorus deficiency. It is worth noting that transgenic plants expressing *PHT1-1* performed significantly better than those harboring and expressing the *PTH1-3* gene homolog under P stress, which may indicate they are involved in different processes and play different roles in phosphorus stress response.

## 4. Discussion

Blueberry root mycorrhizal fungal species are generally diverse and mainly belong to either ascomycetes and/or basidiomycetes phyla. Typical mycorrhizal fungal genera associated with blueberries include *Aspergillus*, *Penicillium*, *Fusarium*, and *Oidiodendron* [16,46]. Reported studies show that endomycorrhizal fungi are distinctive and specific to the blueberry cultivar, the soil environment, geographical location, and/or the time of sampling [47]. In this study, 13 of the isolated fungal strains were identified as ascomycetes. ITS DNA sequencing confirmed that most of these belonged to *Penicillium* and *Fusarium* genera (Figure 1B). Morphologically, isolated endomycorrhizal fungi were mainly white to off-white in color with flocculent or tapetum hyphae, and individual colonies were noted to secrete pink (which was characteristic of *Fusarium* like A4), yellow, or green pigment (Figure 1A), and basically in agreement with previous reports [46]. 

The growth of blueberry plants improved after inoculation with beneficial endomycorrhizal fungi with corresponding changes in phenotype and physiological indicators. Blueberries naturally form ERM with indigenous endomycorrhizal fungi in commercial blueberry nurseries [48]. However, the rate of root fungal colonization is typically low, with 10–16% reported within the first year [20,49]. Our study demonstrated significant increases in infection rate (up to 68%) following inoculation with the endomycorrhizal fungi isolates (Figure 3B), exceeding previous reports of c.a. 30% infectivity [20,50]. Accordingly, inoculation with endomycorrhizal fungi isolates and successful establishment of ERM significantly improved root vigor of plantlets (Figure 3C) and led to robust development—both plant height and crown growth significantly increased (Figure 2). This is consistent with reported root vigor enhancement of ryegrass and apple trees following inoculation with beneficial endomycorrhizal fungi [51,52]. In this study, we demonstrated that this increase in blueberry root vigor promoted nutrient absorption and accumulation—particularly nitrogen and phosphorus in the roots, stems, and leaves (Figure 4)—which is in keeping with other reported studies [53]. Interestingly, the results also show that mycorrhizal fungi inoculation had a better growth-promoting effect on blueberries than commercial bacterial fertilizer, reinforcing their potential commercial value in blueberry cultivation. 

Until now, the molecular basis of ERM fungi and blueberry plant symbiosis had not been reported. Although *VcPHT1s* gene expression generally increased in blueberries, our study shows that the amounts vary and that it is fungal-strain dependent. Inoculating blueberry seedlings with beneficial endomycorrhizal fungi “upregulated” *VcPHT1s* expression (Figure 5) within the roots, leading to noted P accumulation within the plant (Figure 4). We reason that the ERM structure expanded the P absorption capacity of the roots, thereby increasing the bioavailability of rhizosphere P and accordingly eliciting activity from the relevant genes [54,55]. However, expression of *PHT1-2*, *3*, and *4* in blueberry roots declined significantly after inoculation with isolate A4—in contrast to other isolates but consistent with reported studies on non-blueberry plants [5,56]. This is probably because plants associated with mycorrhiza can absorb phosphorus in the rhizosphere either directly or indirectly. In plants other than blueberries, most of the phosphate transported via the mycorrhizal pathway leads to downregulation of *VcPHT1s* genes in the root system because direct root absorption is partially inhibited [57]. For example, in studies of soybeans and wheat inoculated with mycorrhizal fungi, the expression of *PHT1s* in roots was mostly downregulated to differing extents [57,58]. However, this probably doesn’t apply to blueberry roots because they only have a small biomass and don’t possess root hairs [3]. We believe that the absorption capacity of ERM-blueberry root structures is relatively limited and that the amount of transported phosphate is inadequate for standard growth. Hence, most *VcPHT1s* in the root zone are upregulated for direct P absorption from surrounding soil to meet these requirements.

Generally, expression levels and function of *VcPHT1s* gene products in roots differed from those observed in blueberry leaves following inoculation with endomycorrhizal fungi (Figure 4 and Figure 5). The *PHT1-1* gene was expressed at significantly higher levels in roots than in leaves, while the converse was noted for *PHT1-2* expression. This suggests that regulation of *PHT1s* gene expression and/or post-translational modulation—such as phosphate transporter traffic facilitators (PHF)—may differ between plant organs [59,60]. 

Cloning and overexpression of *VcPHT1s* genes into *A. thaliana* confirmed their biological function under restricted P conditions. Study results demonstrated that *PHT1-1* and *PHT1-3* aided *A. thaliana* to use P more effectively and maintain requisite levels for effective plant growth under P stress conditions (Figure 7). Similar findings have been reported in tomatoes with *LePT1* and *LePT2* [11]. Wang et al. (2013) also reported that overexpression of an analogous *PHT1* gene (*TaPHT1.2*) in *A. thaliana* increased the number of root tips and the length of lateral roots to promote P absorption under limited P conditions [61]. Previous studies have likewise shown that the number of tillers in transgenic rice endowed with tomato *SIMPT3;1* was significantly higher than the wild-type plant and that its capacity for P adsorption had significantly improved [62]. These studies support the view that the P transporter gene facilitates better absorption of external P by promoting plant root growth. However, the specific role of *PHT1-1*, *PHT1-2*, *PHT1-3*, and *PHT1-4* in the transport and absorption of phosphorus in blueberries requires further investigation.

## 5. Conclusions

This is the first reported work detailing the isolation and characterization of ERM fungi from blueberries with phenotypic and genetic evidence, including confirmation of their beneficial effects (especially isolates X2, X6, X11, X24, A2, and A4) in the growth of plant height and crown, root activity, absorption of N and P. Two of the isolates (X5 and X10) were unclassified and are subject to further characterization. The outcomes of this work serves as a solid foundation for future endomycorrhizal fungi research and development—discovery of new isolates and their growth-promoting activities—and their application in expanding the blueberry industry in southern China and beyond.

## Figures and Tables

**Figure 1 jof-07-00584-f001:**
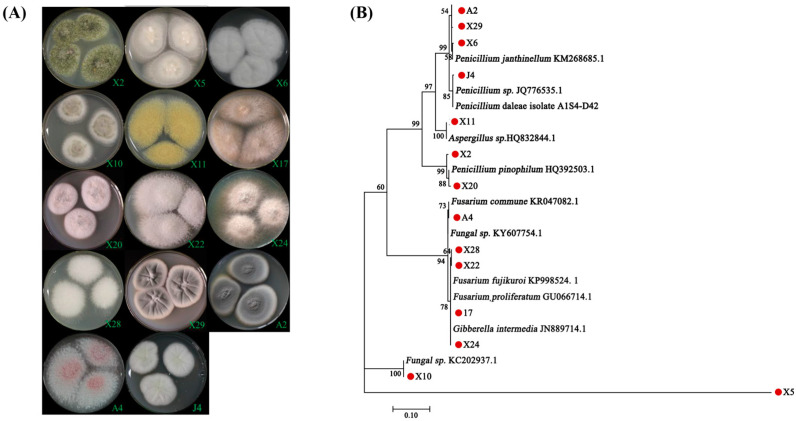
Colony features and phylogenetic tree of ITS sequences derived from endomycorrhizal fungi isolated from blueberry plant roots. (**A**) X2, X5, X6, X10, X11, X17, X20, X22, X24, X28, and X29 were isolated from the blueberry variety “Sharpblue”. A2 and A4 were isolated from the “O’Neal” variety. J4 was isolated from the variety known as “Premier”. (**B**) Phylogenetic analysis was performed by the MEGA software (version 7.0) using the maximum-likelihood algorithm. Numbers in the figure represent bootstrap values (1000 replicates).

**Figure 2 jof-07-00584-f002:**
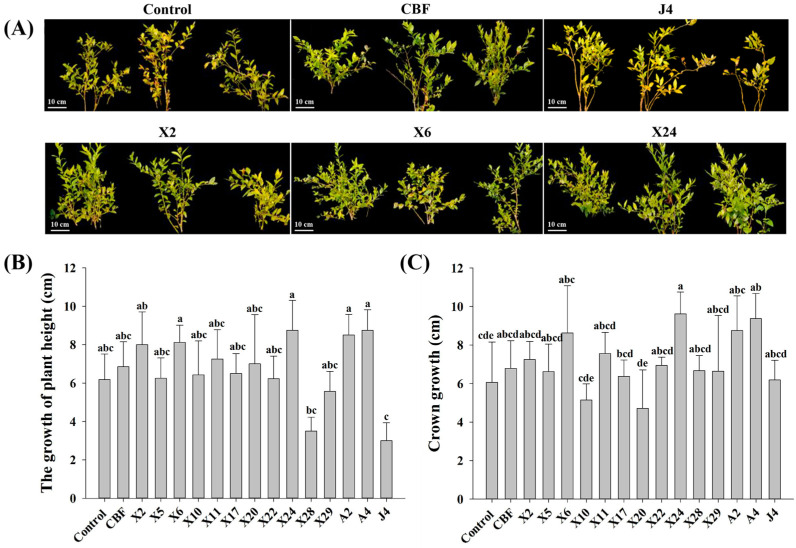
Effects of inoculation with endomycorrhizal fungi on blueberry (**A**) phenotype, (**B**) plant height, and (**C**) crown of blueberries. Representative phenotypes are shown in Figure 2A. Control plants were treated with conventional organic fertilizer with nutrient solution, CBF was treated with commercial bacterial fertilizer, and the remaining treatments were treated with conventional organic fertilizer with nutrient solution, then inoculated with endomycorrhizal fungi isolates. Columns marked with different lowercase letters indicate significant differences (*p* < 0.05) between genotypes. Error bars represent the standard error of means (*n* = 3).

**Figure 3 jof-07-00584-f003:**
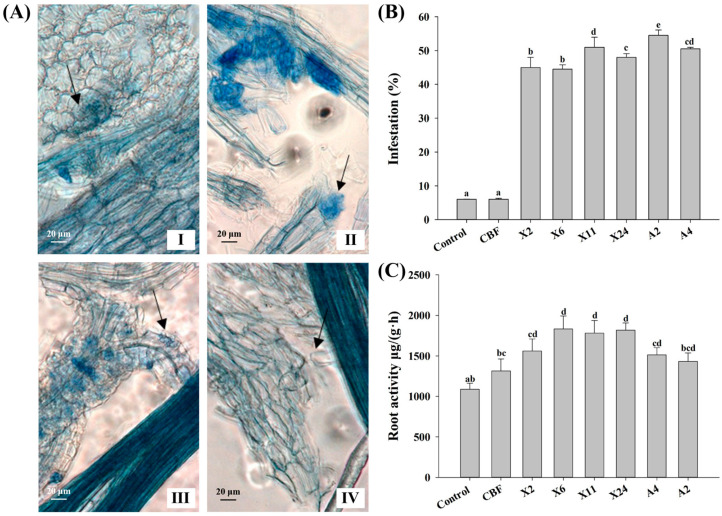
Infestation status and root activities of the seedlings after inoculation with endomycorrhizal fungi. (**A**) Pictograph of fungal colonization morphology in blueberry roots under 100× magnification, depicting hyphal structure: (I) intracellular mycelial coils; (II) intracellular mass; (III) intercellular spread via gaps between cells; and (IV) longitudinally or laterally intercellular and intracellular extension. The black arrow points to the mycelial structure. (**B**) Infection rate of blueberry seedlings after inoculation with different fungi. (**C**) Effect of fungal inoculation on blueberry root activity. Control plants were treated with conventional organic fertilizer with nutrient solution, CBF was treated with commercial bacterial fertilizer, and the remaining treatments were treated with conventional organic fertilizer with nutrient solution, then inoculated with beneficial endomycorrhizal fungi isolates. Columns marked with different lowercase letters indicate significant differences (*p* < 0.05) between genotypes. Error bars represent the standard error of means (*n* = 3).

**Figure 4 jof-07-00584-f004:**
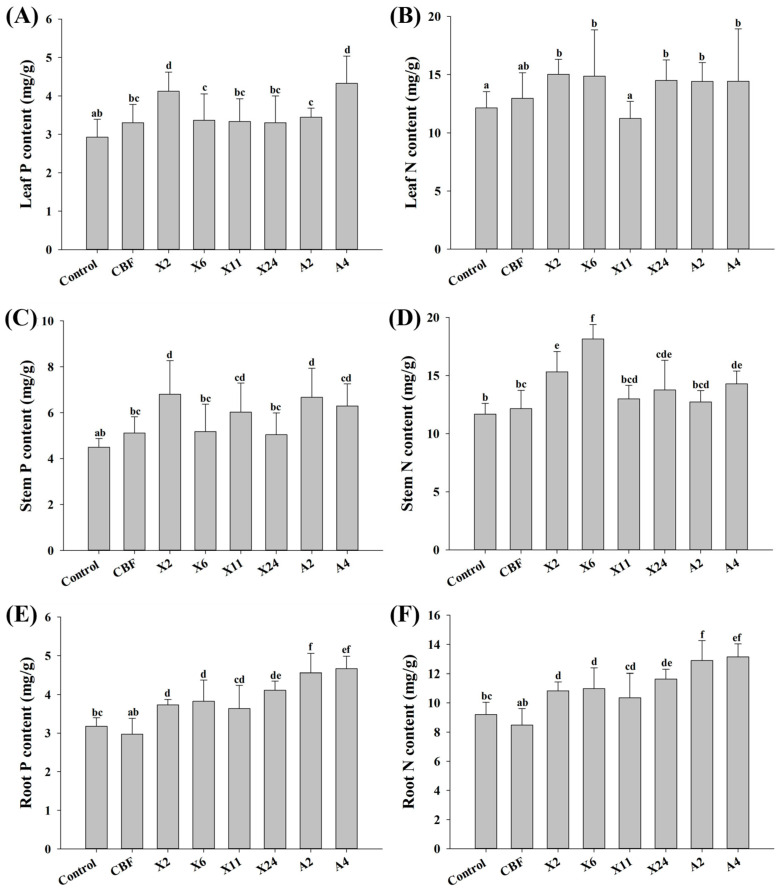
Phosphorus content in (**A**) leaves, (**C**) stems, and (**E**) roots; and nitrogen content in (**B**) leaves, (**D**) stems, and (**F**) roots of blueberry seedlings under different treatment conditions. Control plants were treated with conventional organic fertilizer with nutrient solution, CBF was treated with commercial bacterial fertilizer, and the remaining treatments were treated with conventional organic fertilizer with nutrient solution, then inoculated with beneficial endomycorrhizal fungi isolates. Columns marked with different lowercase letters indicate significant differences (*p* < 0.05) between genotypes. Error bars represent the standard error of means (*n* = 3).

**Figure 5 jof-07-00584-f005:**
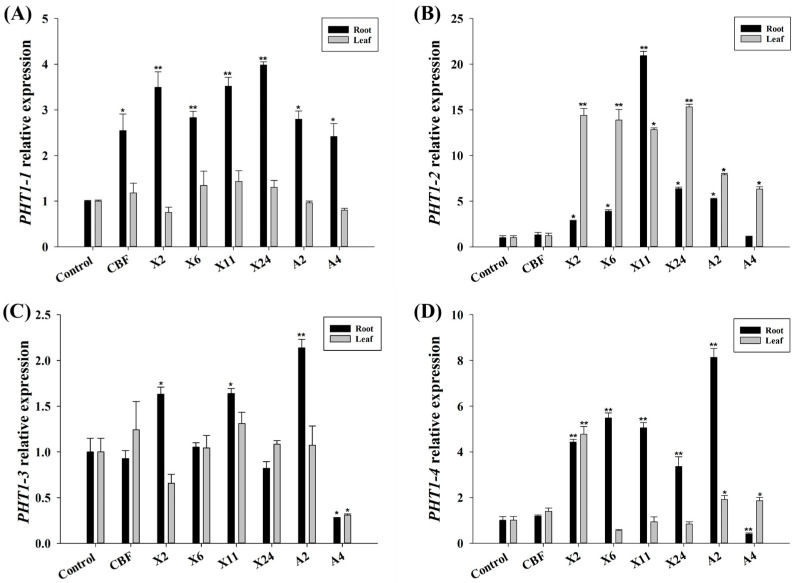
Effect of selected fungi inoculations on the relative expression levels of (**A**) *PHT1-1*, (**B**) *PHT1-2*, (**C**) *PHT1-3*, and (**D**) *PHT1-4* genes in blueberry seedling roots and leaves. Control plants were treated with conventional organic fertilizer with nutrient solution, CBF was treated with commercial bacterial fertilizer, and the remaining treatments were treated with conventional organic fertilizer with nutrient solution, then inoculated with beneficial endomycorrhizal fungi isolates. Asterisks indicate significant differences between groups (* *p* < 0.05, ** *p* < 0.01). Error bars represent the standard error of means (*n* = 3).

**Figure 6 jof-07-00584-f006:**
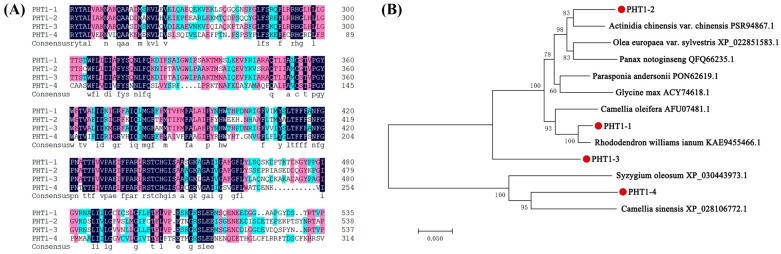
Multiple alignments of amino acid sequences (**A**) and phylogenetic tree (**B**) of the *PHT1* protein family. Identical amino acid residues are shaded in black and similar residues in color. Phylogenetic analysis was performed by the MEGA software (version 7.0) using the neighbor-joining algorithm. Numbers in the figure represent bootstrap values (1000 replicates).

**Figure 7 jof-07-00584-f007:**
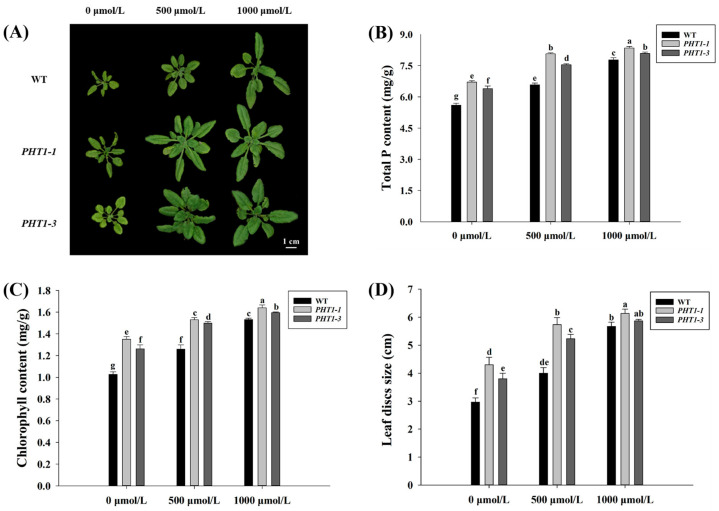
Growth status of *wt* and *PTH1s* transgenic Arabidopsis plants, including (**A**) phenotype, (**B**) total P content, (**C**) chlorophyll content, and (**D**) leaf disc size. The treatments included: zero P treatment (0 μmol/L), low P treatment (500 μmol/L), and normal P treatment (1000 μmol/L). Columns marked with different lowercase letters indicate significant differences (*p* < 0.05) between genotypes. Error bars represent the standard error of means (*n* = 3).

**Table 1 jof-07-00584-t001:** Modified Hoagland nutrient solution formula.

Reagent	Concentration (g/L)	Reagent	Concentration (g/L)
K_2_HPO_4_	0.140	H_3_BO_3_	2.86
MgSO_4_	0.490	MnSO_4_	1.81
Ca(NO_3_)_2_·4H_2_O	1.180	ZnSO_4_	0.22
KNO_3_	0.150	Na_2_MoO_4_·2H_2_O	0.03
FeSO_4_·7H_2_O	0.011	CuSO_4_·5H_2_O	0.08
EDTA·2Na	0.015		

**Table 2 jof-07-00584-t002:** Sequences of qRT-PCR primers for *PHT1-1*, *PHT1-2*, *PHT1-3*, and *PHT1-4*.

Gene Name	Sequence (5′→3′)-Forward	Sequence (5′→3′)-Reverse
*PHT1-1*	CTGGTTCACCGTAGCACTTATC	GAGTCCAGTGGTTGTATGGAATAG
*PHT1-2*	CCGACTACGTTTGGAGGATTAT	AGTGTATCGGGCAGTTTCAG
*PHT1-3*	CCCGTTACACTGCCCTTATC	GGCCTCTTCGGGTTCTATTT
*PHT1-4*	CGCTTTACTAGGAGCCGTAATC	ATAGAGAATCCGCAACCGAAC

**Table 3 jof-07-00584-t003:** Sequences of full-length primers for *VcPHT1s* genes.

Gene Name	Sequence (5′→3′)-Forward	Sequence (5′→3′)-Reverse
*PHT1-1*	GGAATTCCCAGCTATGGCCAAAGAACAATTACA	GCTCTAGAGCGAAATAGCGTAAGCATAAAAAAGTA
*PHT1-2*	GGAATTCCAGAAAGAAGTGTTGAGAAAGGGAGA	CGGGATCCGGATTCGAAAAGTACAGGACCTCTA
*PHT1-3*	GGAATTCCATGCCTAGAGAACAGTT	CGGGATCCGTTAAACCGGAGCGGTC
*PHT1-4*	CGGGATCCCGTTCACAAAAGGAAA	GCTCTAGAGCTACAAATTAATAGT

## Data Availability

Not applicable.

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
