# Peer review of "Isolation and Characterization of Endomycorrhizal Fungi Associated with Growth Promotion of Blueberry Plants"

_jof, 2021, doi:10.3390/jof7080584_

Round 1

Reviewer 1 Report

Notes about the manuscript

Isolation and Characterization of Endomycorrhizal Fungi Associated with Growth Promotion of Blueberry Plants

Line 73 - Scagel (2005) is cited directly, but the rules are to cite a number at the end. This was done, so it is unnecessary to repeat.

Line 75 – Rhododendron is a genus so please use italics.

Line 88 - … relative ease...” I don’t understand… ease of cultivation?

Line 104 - mycorrhizal endomycorrhizal fungi ... just “endomycorrhizal fungi” would be enough.

Line 295 – could it be possible that the negative effects are results of a pathogenic fungus isolated, since at last X28 is a Fusarium and many Fusarium species are pathogenic to plants with similar symptoms? OJNly one Fusarium (A4) resulted in good answer, from 4 isolated. This is not discussed in the text.

Line 444 – the pink pigment of the Fusarium mycelium shown specially in Figure 1, as for example in A4, is also characteristic and should be mentioned here.

Author Response

Dear Reviewers

We are thankful for having the opportunity to revise the manuscript. We also express our sincere gratitude to you, and the anonymous reviewers for the constructive suggestions and the proposed corrections which enable us to improve the quality of the manuscript (jof-1293123), therefore, to disseminate our work at the highest possible quality.

We have considered all the issues mentioned in the reviewers' comments carefully, and revised the manuscript accordingly, where appropriate. We outlined every change made in response to their comments, and prepared a detailed, point-by-point response, which is given in blue text after each of the comments from the reviewers. In the revised manuscript, we have also highlighted the changes are as red text.

We believe that the revised manuscript has been improved considerably for further consideration by Journal of Fungi.

We would like to thank you again for your kind consideration.

Response to the comments for the Reviewers

Reviewer #1

1. Line 73 - Scagel (2005) is cited directly, but the rules are to cite a number at the end. This was done, so it is unnecessary to repeat.

Response: Thank you for your suggestion. We have deleted “(2005)” in Line 74 of the text.

2. Line 75 - Rhododendron is a genus so please use italics.

Response: Thanks for the correction. We have corrected it, see Line 75.

3. Line 88 - … relative ease...” I don’t understand… ease of cultivation?

Response: We replaced “relative ease” with “relative ease of use”; see Line 88.

4. Line 104 - mycorrhizal endomycorrhizal fungi ... just “endomycorrhizal fungi” would be enough.

Response: Thanks for the correction. We have deleted “mycorrhizal”; see Line 104.

5. Line 295 - could it be possible that the negative effects are results of a pathogenic fungus isolated, since at last X28 is a Fusarium and many Fusarium species are pathogenic to plants with similar symptoms? OJNly one Fusarium (A4) resulted in good answer, from 4 isolated. This is not discussed in the text.

Response: Thank you for your advice. After careful consideration, we give the following reply to your question: first of all, the results of this fungal isolation experiment and the previous classification of blueberry endomycorrhizal fungi do not reflect the general promotion and inhibition of the genus of fungi. Therefore, in the text, we have not discussed whether the growth inhibition of X28 is due to the pathogenicity of Fusarium oxysporum. Secondly, this paper focuses more on the growth-promoting effect of blueberry endomycorrhizal fungi, rather than the pathological reasons of blueberry endomycorrhizal fungi inhibiting growth. We hope to keep the focus unchanged, but we have taken your suggestion as a consideration for future research. Thank you very much.

6. Line 444 - the pink pigment of the Fusarium mycelium shown specially in Figure 1, as for example in A4, is also characteristic and should be mentioned here.Response: Thanks. We have added this part of the content to results and discussion corresponding section (Lines 266, 446 and 447).

Sincerely yours,

Wenrong Chen

Yunying Fang

Reviewer 2 Report

I think the work is interesting for people who work in the relationship plant microorganisms. Some of my comments regarding the document are detailed below 

In line 6 of the document I think it would be more appropriate to speak of mutualism than of symbiosis, since mutualism is a relationship of mutual benefit and includes symbiosis as a closer relationship and of co-dependency. 

Line 68 is missing a reference 

In line 90 the term demonstrate is used, I would change this to evaluate, since it is in the introductory section and I believe that the demonstration derives from the discussion of the results. 

In line 142 it is indicated that a NJ analysis was used, currently there are more robust algorithms, I think it could be evaluated by another analysis. NJ might be enough, I think you should rate this based on the other reviewers' comments

In the paragraph between lines 253 to 255, I think it is necessary to review the statistical analysis, since although it is far from my expertise, the analysis of variance requires a normal distribution and homoscedasticity. In addition, the distribution analysis needs a number of replicates higher than what you provide in the document, as an alternative, it could be evaluated by an equivalent non-parametric test.  

In figure 1B (near 273) the phylogenetic analysis has no scale in the length of the branches 

In Figures 2 3 4 5 and 7, the error bars correspond to the standard error. Normally the standard deviation is used as a measure of dispersion of the data since the standard error tends to 0 as the number of samples increases.

in line 506 507 "including confirmation of their beneficial effects", I think it could be changed to "confirmation of their beneficial effects in ... (measured parameters)", as there could be more beneficial parameters or some untested detrimental effects. 

Author Response

Dear  Reviewers

We are thankful for having the opportunity to revise the manuscript. We also express our sincere gratitude to you, and the anonymous reviewers for the constructive suggestions and the proposed corrections which enable us to improve the quality of the manuscript (jof-1293123), therefore, to disseminate our work at the highest possible quality.

We have considered all the issues mentioned in the reviewers' comments carefully, and revised the manuscript accordingly, where appropriate. We outlined every change made in response to their comments, and prepared a detailed, point-by-point response, which is given in blue text after each of the comments from the reviewers. In the revised manuscript, we have also highlighted the changes are as red text.

We believe that the revised manuscript has been improved considerably for further consideration by Journal of Fungi.

We would like to thank you again for your kind consideration.

Response to the comments for the Reviewers

Reviewer #2

Reviewer #2: I think the work is interesting for people who work in the relationship plant microorganisms. Some of my comments regarding the document are detailed below. 

Response: It’s cheerful for us to hear affirmation of our research. We are grateful to you for reviewing our manuscript, and the valuable comments provided, which helped us improve our revised manuscript greatly. We have carefully considered and addressed the comments raised below.

1. In line 6 of the document I think it would be more appropriate to speak of mutualism than of symbiosis, since mutualism is a relationship of mutual benefit and includes symbiosis as a closer relationship and of co-dependency.

Response: Thanks for the advice. After consideration, we have replaced “symbiosis” with “mutualism” in Lines 14 and 46 to emphasize the closer relationship between plants and endomycorrhizal fungi.

2. Line 68 is missing a reference.

Response: Thanks for the correction. We have corrected it, see Lines 68 and 69.

3. In line 90 the term demonstrate is used, I would change this to evaluate, since it is in the introductory section and I believe that the demonstration derives from the discussion of the results.

Response: Thanks for the correction. We have replaced “evaluate” with “demonstrate” in Line 90.

4. In line 142 it is indicated that a NJ analysis was used, currently there are more robust algorithms, I think it could be evaluated by another analysis. NJ might be enough, I think you should rate this based on the other reviewers' comments.

Response: Thanks for your suggestion. We have used the maximum likelihood method to reconstruct the phylogenetic trees for evaluation, and the following parts have been modified (Figure 1B, Lines 143, 272 and 279).

5. In the paragraph between lines 253 to 255, I think it is necessary to review the statistical analysis, since although it is far from my expertise, the analysis of variance requires a normal distribution and homoscedasticity. In addition, the distribution analysis needs a number of replicates higher than what you provide in the document, as an alternative, it could be evaluated by an equivalent non-parametric test.

Response: Thank for the comment. We have added the sentence “Data were log-transformed to achieve normality where needed.” (Lines 255-256) in the revised maunuscript). We checked the nonmality of data distribution before one-way ANOVA analysis. The data were either normal distributed or being log-transformed if they were not. We recken it is not necessary to conduct non-prametric test.

6. In figure 1B (near 273) the phylogenetic analysis has no scale in the length of the branches.

Response: Thanks for the correction. We have reconstructed the phylogenetic trees with scale in the length of the branches (Figure 1B).

7. In Figures 2 3 4 5 and 7, the error bars correspond to the standard error. Normally the standard deviation is used as a measure of dispersion of the data since the standard error tends to 0 as the number of samples increases.

Response: We disagree and consider standard error of mean (SE) appropriate. SE, as widely used as SD or confidence interval, is calculated from standard deviation (SD) as below. SE provide a measure of how far the sample mean is from the population mean. As you have pointed out earlier, our data sample size is small and appropriate under these circumstances.

n = sample size.

In our case where n = 3, SD = 1.73 × SE. We reckon SE is equally effective and easy to understand as SD.

8. In line 506 507 "including confirmation of their beneficial effects", I think it could be changed to "confirmation of their beneficial effects in ... (measured parameters)", as there could be more beneficial parameters or some untested detrimental effects.

Response: Thanks for your helpful advice. We have replaced “including confirmation of their beneficial effects” with “including confirmation of their beneficial effects (especially isolates X2, X6, X11, X24, A2 and A4) in the growth of plant height and crown, root activity, absorption of N and P, and so on” (Lines 510-511).

Sincerely yours,

Wenrong Chen

Yunying Fang